# An Efficient End-to-End Framework for Localized Second-Order Pooling

**Zhongyan Zhang**     *zy.franky.lol@gmail.com*
*School of Computing and Information Technology, University of Wollongong Australia*

**Saimunur Rahman**     *saimun.rahman@csiro.au*
*Data61, CSIRO*

**Piotr Koniusz**     *piotr.koniusz@unsw.edu.au*
*The University of New South Wales (UNSW) and Data61♥CSIRO*

**Luping Zhou**     *luping.zhou@sydney.edu.au*
*School of Electrical and Computer Engineering, The University of Sydney, Australia*

**Lei Wang**     *leiw@uow.edu.au*
*School of Computing and Information Technology, University of Wollongong Australia*

**Reviewed on OpenReview:** *https://openreview.net/pdf?id=nJBeXjxAqx*

## Abstract

Second-order pooling has proven effective for deep image classification by representing an image with the covariance matrix of its local feature descriptors. However, the diverse visual content in images leads to local descriptors being distributed as multiple modes in the feature space, limiting the effectiveness of a single global covariance matrix. In this work, we propose an efficient end-to-end framework for localized second-order pooling. Our approach jointly learns clusters of local feature descriptors across the entire training set, adaptively assigns the descriptors of each image to the appropriate clusters, computes a localized covariance matrix based on the descriptors assigned to each cluster, and then integrates these matrices to form the image representation.This is achieved through an attention-based local cluster mining branch that automatically identifies the clusters that each local descriptor shall be assigned to. Furthermore, to manage the significant computational overhead incurred by the use of multiple local covariance matrices, we utilize a simple but efficient sample-adaptive feature fusion scheme. This scheme adaptively generates fusion weights for each localized covariance matrix using a lightweight predictor, ensuring both computational efficiency and flexibility. Extensive experiments on multiple fine-grained and large-scale image classification datasets demonstrate that our method consistently improves performance when integrated with state-of-the-art second-order pooling methods and leading network architectures. Ablation studies further verify the efficiency of our feature fusion scheme compared to the existing common alternatives.

## 1 Introduction

With the widespread adoption of deep neural networks, an image is typically transformed—via a series of layer-by-layer projections—into an $h \times w \times d$ feature map, consisting of $hw$ local feature descriptors, each with $d$ channels. To derive a global image representation from this feature map, pooling operations are commonly

applied. While sum-pooling and max-pooling capture only the first-order statistics of local descriptors, second-order pooling goes a step further by extracting covariance information among the channels (i.e., the dimensions of the local descriptors). This process produces a $d \times d$ covariance matrix, which, after appropriate processing, is reshaped into a long feature vector to represent the image. Second-order pooling has demonstrated strong performance across a range of computer vision tasks (Sun et al., 2021; Zhang et al., 2020; Rahman & Moghadam, 2024), with fine-grained classification (Li et al., 2018; Rahman et al., 2023) often cited as a representative application.

Most existing second-order pooling methods (Li et al., 2018; Rahman et al., 2020; Engin et al., 2018; Dai et al., 2022; Rahman et al., 2023; Gao et al., 2019) compute a single global covariance matrix over all local descriptors extracted from an image. This practice implicitly assumes that the distribution of these local descriptors in the feature space is unimodal. However, given the rich and diverse visual content present in real-world images, this assumption rarely holds, and a single global covariance matrix may fail to accurately capture the complex second-order statistics needed for effective representation.

To overcome this limitation, some studies have attempted localized second-order pooling strategies (Li et al., 2017b; Wang et al., 2018). They either apply offline clustering to fixed deep features or replace clustering with multiple projection layers. The former breaks the interaction between clustering and feature learning, while the latter often yields redundant or misaligned subspaces. Although effective to some extent, these approaches do not provide a unified, end-to-end framework for learning descriptors, discovering latent structure, and computing localized second-order representations.

To improve this situation, we propose an efficient end-to-end framework for localized second-order pooling, which computes covariance matrices within the clusters that are jointly and explicitly discovered during network training. Specifically, we introduce an attention-based local cluster mining branch into the deep network, which is jointly trained with the network on the entire dataset. This branch automatically identifies the underlying clusters from training data, encouraging local descriptors with similar visual patterns to be grouped together as they are learned. To represent an image, instead of using all local descriptors from this image to compute a single global covariance matrix, the proposed network adaptively assigns the descriptors of this image to the appropriate clusters, computes a localized covariance matrix based on the descriptors assigned to each cluster. This online clustering process effectively partitions the feature space into multiple regions, each corresponding to semantically coherent groups. Within each region, second-order statistics can be captured more accurately by the corresponding covariance matrix.

Furthermore, we address a direct consequence of localized approach, that is, the significant computational overhead arising from the increased number of covariance matrices. In practice, each $d \times d$ covariance matrix is typically reshaped into a feature vector of dimension $d(d + 1)/2$ by retaining only the upper triangular part due to symmetry. Simply concatenating all local covariance features (Li et al., 2017b) greatly increases dimensionality and model parameters. To address the computational overhead, we propose a sample-adaptive feature fusion scheme that dynamically aggregates local covariance features. Instead of relying on fixed or global weighting, our method learns dynamic, instance-specific fusion weights that model the contribution of each of the localized covariance matrices to the final class prediction, resulting in a more compact yet discriminative representation.

Together, these components constitute an efficient end-to-end framework for localized second-order feature pooling. Our framework not only dynamically captures complex channel-wise relationships but also effectively manages computational costs. Furthermore, experimental results demonstrate that our approach integrates well with various deep neural network architectures and enhances state-of-the-art second-order pooling methods across both fine-grained and large-scale classification benchmarks.

Our contributions are summarized as follows:

1. We propose an efficient framework for localized second-order pooling in an end-to-end manner. This framework jointly learns feature descriptors and identifies their clusters across the entire training set, facilitating more effective second-order pooling for feature descriptors.

2. To mitigate the increase on computational costs, we develop a sample-adaptive feature fusion scheme. This scheme flexibly merges features from all local covariance matrices per image, not only helping to improve performance but also better reducing computational overhead compared to other fusion alternatives.

3. We perform extensive experimental studies on both fine-grained and large-scale image classification datasets, validating the efficacy of the proposed framework.

## 2 Related work

Second-order pooling has long been recognized as a powerful tool for capturing pairwise correlations among feature components. Initially employed in classical vision tasks such as object classification and tracking (Tuzel et al., 2006; 2007), covariance matrices served as standalone region descriptors. With the advent of deep learning, they have been integrated into neural networks as trainable modules. Pioneering works such as Bilinear CNN (Lin et al., 2015) and DeepO$_2$P (Ionescu et al., 2015) laid the foundation for end-to-end second-order pooling. Subsequent efforts addressed the challenges posed by symmetric positive definite (SPD) matrices—namely their high dimensionality and non-Euclidean geometry—through techniques such as matrix normalization (Li et al., 2018; Koniusz et al., 2018; Gao et al., 2019; Wang et al., 2021; Song et al., 2021; 2023a;b), compact and low-rank pooling (Gao et al., 2016; Kong & Fowlkes, 2017; Zheng et al., 2019; Rahman et al., 2020; Yu et al., 2020; 2021; Wang et al., 2022; Yu et al., 2022b), linear transformations (Brooks et al., 2019; Huang & Gool, 2017; Nguyen et al., 2019; Yu & Salzmann, 2017; Wang et al., 2021), and inverse covariance (Rahman et al., 2023).

**Global Second-Order Pooling Approaches.** Among these advances, global second-order pooling approaches compute a single covariance matrix over all local descriptors from an image, simplifying the representation pipeline while retaining expressive second-order statistics. Notable examples include matrix square-root normalization (Li et al., 2018) and compact bilinear pooling (Gao et al., 2016), which improve representation quality through power normalization and dimensionality reduction. Kernel pooling methods (Cui et al., 2017) further enrich the representation by projecting features into high-dimensional Hilbert spaces. Despite their effectiveness, these global pooling techniques inherently assume feature homogeneity and often dilute critical local variations, limiting their capacity to capture fine-grained distinctions in complex visual scenes. As a result, the performance may be suboptimal in tasks that require detailed spatial discrimination.

**Local Second-Order Pooling Approaches.** To address the global approach's limitations, localized pooling methods compute multiple covariance matrices for distinct feature regions or in different subspaces, offering more nuanced representations. High-order local pooling (Li et al., 2017b) encodes local descriptors as Gaussian components based on a pre-learned dictionary, where feature grouping and descriptor extraction are conducted in a decoupled manner. In contrast, GM-SOP (Wang et al., 2018) learns a set of nonlinear projection layers that implicitly map local features into multiple subspaces, where separate local covariance matrices are computed. Although these methods enhance representation fidelity, there is still room for further improvement and exploration. High-order local pooling decouples clustering from descriptor learning, relying on offline processes, while GM-SOP does not explore the potential of an explicit clustering mechanism within its end-to-end learning framework. In contrast, our proposed framework integrates an attention-driven cluster mining mechanism directly within the network, enabling explicit feature space partition for localized second-order pooling in an end-to-end training manner.

In addition, the feature fusion scheme in GM-SOP remains limited, as the softmax-normalized weighted sum restricts the final representation to a convex combination of covariance experts, often leading to over-smoothed second-order features. Therefore, the method relies on an additional diversity regularizer to prevent the experts from collapsing into similar subspaces, indicating that the fusion itself provides limited incentive for learning truly distinct structures.

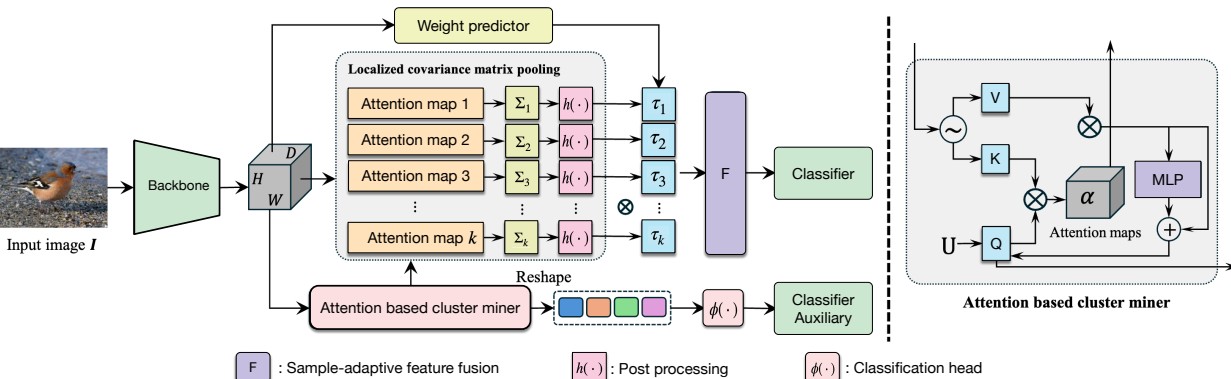

Figure 1: Overview of the proposed end-to-end localized second-order pooling framework (LSOP), including the attention-based cluster miner, localized covariance computation, sample-adaptive weight predictor, and feature fusion.

## 3 Proposed method

Our method consists of three main components: 1) an attention-based local cluster mining branch, which groups local descriptors into visually similar clusters; 2) a localized covariance matrix branch, which assigns the local descriptors of an input image and computes a covariance matrix for each group, followed by post-processing steps such as normalization and vectorization; and 3) a sample-adaptive feature fusion scheme, which generates fusion weights to aggregate the localized covariance features for final classification. The overall pipeline is illustrated in Figure 1. The following subsections describe each component in detail.

### 3.1 Attention-based Cluster Mining

The goal of our online clustering component is to perform clustering and feature learning simultaneously during training, while maintaining a consistent ordering of clusters across images so that downstream fusion weights can be reliably applied to the localized covariance features. This requires a differentiable, data-adaptive procedure that updates cluster assignments in real time and yields stable, repeatable indices. We considered several alternatives. VQ-based approaches (van den Oord et al., 2017; Esser et al., 2021) use hard assignments to a static codebook, which can restrict gradient flow and make the ordering of codes across images less stable. DeepCluster-style methods (Caron et al., 2018; Zhan et al., 2020) typically rely on separate or batch-level clustering steps (e.g., k-means) and thus decouple assignment from the main optimization, often leading to label churn and additional synchronization overhead. In contrast, we adopt a slot attention scheme (Locatello et al., 2020), which performs soft, differentiable assignments and updates the slot prototypes jointly with the network. Recent work has noted its connection to k-means (Yu et al., 2022a), while retaining the benefit that cluster updates are directly driven by the task loss. This yields an online procedure with stable slot indexing across images and efficient end-to-end optimization, which is well matched to our setting.

**Slot-attention Module.** The basic idea of slot attention is to embed a set of $m$ input vectors into a set of $K$ output vectors that refer to as the slots in an iterative way. In our case, the input vectors are the $hw$ local descriptors from the feature map $\mathbf{F} \in \mathbb{R}^{d \times hw}$ and the output vectors are the $K$ learnable slots $\mathbf{U} \in \mathbb{R}^{d \times K}$. At each iteration, three learnable linear transformations $\mathbf{q}$, $\mathbf{k}$, and $\mathbf{v}$ are used to map inputs and slots to a common dimension $d$ and a cross-attention operation is conducted followed by a residual network. In particular, the slots $\mathbf{U}^{t+1}$ at the $(t+1)$th iteration can be derived from the previous ones $\mathbf{U}^t$ with the following forms:

$$\hat{\boldsymbol{\alpha}} = \text{softmax}_k \frac{\mathbf{q}(\mathbf{U}^t)^\top \cdot \mathbf{k}(\mathbf{F})}{\sqrt{d}},$$

$$\boldsymbol{\alpha} = [\boldsymbol{\alpha}_1, \boldsymbol{\alpha}_2, \ldots, \boldsymbol{\alpha}_K]^\top, \boldsymbol{\alpha}_k = \frac{\hat{\boldsymbol{\alpha}_k}}{\sum_{hw} \hat{\boldsymbol{\alpha}_{ki}}},$$

$$\mathbf{U}^{t'} = \mathbf{U}^t + \mathbf{v}(\mathbf{F}) \cdot \boldsymbol{\alpha}^\top, \tag{1}$$

$$\mathbf{U}^{t+1} = \mathbf{U}^{t'} + \text{MLP}(\text{LayerNorm}(\mathbf{U}^{t'})),$$

where the subscript $k$ indicates that the softmax is applied along the slot dimension. The above process will be conducted $T$ times to obtain the final slots $\mathbf{U}^T$. In the meantime, the obtained attention map $\boldsymbol{\alpha} \in \mathbb{R}^{K \times hw}$ from the last iteration is used to represent the assignment for each local descriptor, as it denotes their probabilities of being assigned to the slots.

**Module Training as Dictionary-based Encoding.** Having described the internal design of the slot attention block, we now turn to how this module is trained within our framework. In prior works, slot attention is often supervised either through slot-level objectives, such as contrastive losses (Wen et al., 2022; Weinzaepfel et al., 2022) assigned to individual slots, or via unsupervised reconstruction-based losses (Locatello et al., 2020) that encourage slots to decompose an image into distinct parts. However, in our case, supervision is provided solely at the image level through class labels. This setting calls for a training scheme that can effectively guide the slot attention module to learn meaningful and discriminative representations without direct supervision at the slot level. To this end, we adopt a simple yet effective design: after the final iteration of slot refinement, we concatenate all slot embeddings and feed the resulting vector into a lightweight embedding layer followed by a standard classifier trained with cross-entropy loss. This allows each slot to specialize through gradient signals propagated from the global objective, encouraging implicit feature decomposition in an end-to-end fashion.

Specifically, we concatenate the slots of the final iteration, $\mathbf{U}^T \in \mathbb{R}^{d \times K}$, into a vector $\mathbf{u}' \in \mathbb{R}^{dK}$, and apply an embedding layer before feeding them into the auxiliary classifier, which is trained using the following online clustering loss:

$$L_{oc} = -\sum_{i=1}^{N} y_c \cdot \log \left( \frac{\exp\left(\phi(\mathbf{u}'_i) \cdot \omega_c\right)}{\sum_{j=1}^{C} \exp\left(\phi(\mathbf{u}'_i) \cdot \omega_j\right)} \right), \tag{2}$$

where $\phi(\cdot)$ denotes an embedding function consisting of a layer normalization, a ReLU, and a linear layer. Here, $\omega_c$ and $y_c$ represent the classification weight and label for class $c$, respectively. $C$ and $N$ denote the total number of classes and the number of samples in a mini-batch, respectively.

This design is inspired by dictionary-based encoding methods such as VLAD (Jégou et al., 2010) and its variants, which were popular in the Bag-of-Words (BoW) era. These methods assign local descriptors to a set of predefined codewords and aggregate the residuals between descriptors and their assigned centers. The resulting residual vectors are concatenated to form a global image representation for classification. Our slot attention mechanism follows a similar principle. The initial learnable slots $\mathbf{U}^0$ act as cluster centers, and a soft assignment is performed, where each local descriptor contributes to multiple slots based on attention weights. These assignments are then aggregated to update the slots, and this update implicitly resembles a residual connection between the aggregated features and the current slot state. Unlike conventional dictionary-based encodings, our slots are iteratively refined, but the final representation is still formed by concatenating all slot embeddings for classification.

## 3.2 Localized Covariance Matrices

Before computing localized covariance matrices, it is important to clarify that the attention-based cluster mining module only provides soft assignments that assign local descriptors into semantically coherent groups; it does not perform any statistical aggregation or pooling. Instead, the second-order pooling of local descriptors is conducted in each cluster individually, by computing a localized covariance matrix within each cluster.

Recall that we have obtained the attention map $\boldsymbol{\alpha}$ denoting the soft assignments of each local descriptor with respect to the learned clusters. Instead of hard assignment, we use soft assignment here by considering two issues. First, it avoids the situation that the local descriptors from a given image are only assigned to a subset of the $K$ clusters. This situation usually happens when the visual content of the image does not cover the semantic patterns represented by some cluster centers. In other words, if the number of clusters is limited or if certain centers correspond to rare or specific visual structures, and those structures are absent in the image, then none of the local descriptors from that image will be assigned to those centers. In this case, the local covariance matrices corresponding to the empty clusters cannot be calculated. Second, it fully utilizes all the local descriptors from an image to more reliably estimate each local covariance matrix. This helps to mitigate the small sample issue when only using the descriptors hard-assigned to a specific cluster to estimate the covariance matrix therein. The attention map for each cluster is used as weights to compute a weighted covariance matrix $\boldsymbol{\Sigma}_k$ $(k = 1, \cdots, K)$ for the $k$-th cluster, with its $(p, q)$-th entry as

$$\boldsymbol{\Sigma}_k(p, q) = \frac{1}{1 - \|\boldsymbol{\alpha}_k\|_2^2} \sum_{i=1}^{hw} \alpha_{ki}(x_{ip} - \hat{\mu}_{kp})(x_{iq} - \hat{\mu}_{kq}), \tag{3}$$

where $\boldsymbol{\alpha}_k = (\alpha_{k1}, \alpha_{k2}, \cdots, \alpha_{k(hw)})^\top$ and $x_{ip}$ denotes the $p$-th component of the local descriptor $\mathbf{x}_i$ while $\hat{\mu}_{kp}$ is the weighted mean defined as $\hat{\mu}_{kp} = \sum_{i=1}^{hw} \alpha_{ki} x_{ip}$ over all local descriptors from an image. Repeating this for each local cluster produces the $K$ local covariance matrices.

### 3.3 Feature Fusion of Local Covariance Matrices

With the presence of multiple local covariance matrices, a fusion scheme that can lead to more accurate classification while avoiding a substantial increase on computational cost is needed to fuse the features extracted from the matrices.

A straightforward baseline is to directly **concatenate** the $K$ features into a single vector, as done in (Li et al., 2017b). While this approach retains complete information from all branches, it causes a substantial increase in dimensionality[1] and ignores the relative importance or redundancy among features. To address this, one can introduce a **cross-attention** module, allowing each branch to interact with others dynamically. For example, we could treat the feature vector from a global covariance matrix as the *key* and the feature vector from each local matrix as the *query* and *value*. By cross-attention, the $K$ local feature vectors can be combined by a weighted combination. This enables context-aware fusion but comes at the cost of increased parameters and compute, which may be prohibitive in certain settings. A more compact alternative is to predict a set of $K$ normalized weights (via softmax) that reflect the relative importance of each branch, and use them to perform a **weighted sum** of features. This approach is lightweight and interpretable, but it treats branches independently and lacks the ability to model richer interactions. Building on this, a **gated selection mechanism** can be employed to enforce sparsity by selecting a subset of $K - n$ branches, before applying softmax-weighted fusion, as done in GM-SOP (Wang et al., 2018). Although this introduces stronger selection capability, it depends on the design of the gating function and the choice of the sparsity level, i.e., how many branches to retain.

To achieve more efficient fusion, we propose a **sample-adaptive fusion** scheme that avoids discrete selection altogether. Instead of gating, we predict $K$ continuous scaling factors that softly modulate the contribution of each branch. This approach eliminates the need to set hard thresholds or sparsity levels, while still allowing the model to downweight less relevant features and amplify more informative ones, leading to a more flexible and stable fusion.

**Sample-adaptive Feature Fusion.** The basic idea of this fusion scheme is to measure the importance of each bunch of local descriptors that are assigned to the same cluster and use it to rescale the contribution of the features in the classification loss. To achieve that, a weight predictor consisting of a global average pooling followed by a dimension reduction layer is used to measure the scalers $\boldsymbol{\tau} \in \mathbb{R}^K$ for all the matrices as follows:

---

[1]The dimension of a covariance matrix feature is usually $d(d+1)/2$ after the common steps of normalization and reshaping.

$$\boldsymbol{\tau} = \text{sigmoid}(g(\mathbf{F}) \cdot \mathbf{w}), \tag{4}$$

where $\mathbf{F}$ is a $h \times w \times d$ feature map, $g(\cdot)$ is global average pooling, $\mathbf{w} \in \mathbb{R}^{d \times K}$ is the dimension reduction layer, and sigmoid$(\cdot)$ controls the range of the output. After that, $\boldsymbol{\tau}$ is used to re-weigh the covariance matrices, with a carefully designed regularization term imposed on $\boldsymbol{\tau}$. Our loss function is defined as follows:

$$\mathcal{L}_{sa} = -\sum_{i=1}^{N} y_c \cdot \log \left( \frac{\exp\left(\mathbf{f}_i \cdot \omega_c\right)}{\sum_{j=1}^{C} \exp\left(\mathbf{f}_i \cdot \omega_j\right)} \right) + \beta \sum_{i=1}^{N} \sum_{k=1}^{K} \log\left(\frac{1}{\tau_k}\right), \text{where } \mathbf{f}_i = \sum_{k} \frac{h(\boldsymbol{\Sigma}_k)}{\tau_k} \tag{5}$$

$h(\cdot)$ denotes the operations of matrix normalization and reshaping applied to $\boldsymbol{\Sigma}_k$, producing a $(d(d+1)/2)$-dimensional feature vector. $\beta$ is the coefficient to control the penalty on the small $\tau$.

After assembling all three components together, the proposed framework could be trained with image-wise annotations in an end-to-end manner by integrating the two losses in Eq. (2) and Eq. (5) as

$$\mathcal{L}_{total} = \mathcal{L}_{sa} + \lambda \mathcal{L}_{oc}. \tag{6}$$

## 4 Experiment

In this section, we first introduce the datasets, evaluation metrics, and implementation details used in our experimental study. Next, we compare three mainstream covariance matrix-based feature pooling methods with their counterparts enhanced by the proposed approach. After that, the performance on large-scale dataset is reported. We then present a comprehensive comparison of different feature fusion schemes. Finally, we conduct an ablation study to further investigate the properties of the proposed method.

### 4.1 Datasets, Metric, and Implementation

**Dataset.** We evaluate our method on four fine-grained visual categorization (FGVC) datasets, i.e., Birds (Wah et al., 2011), Cars (Krause et al., 2013), Airplane (Maji et al., 2013), and Dogs (Dataset, 2011), one scene dataset, i.e., MIT (Quattoni & Torralba, 2009), and one large-scale visual classification dataset i.e., ImageNet-1K (Deng et al., 2009). For our experiments, we use the protocol of Bilinear CNN (Lin et al., 2015) which is largely followed in the literature for the first five datasets and follow the protocol in their original papers for the last one. Further details of the datasets are provided in the supplementary materials.

**Metric.** We use average classification accuracy as in literature to evaluate and compare our method, e.g., (Lin et al., 2015; Li et al., 2018; Rahman et al., 2020) for the first five datasets, but top-1 and top-5 error rates for ImageNet-1K instead.

**Implementation detail.** We implement our method using PyTorch, which serves as the platform for both our codebase and the pretrained backbone models. For all datasets except ImageNet-1K, we adopt backbones initialized with weights pretrained on the ImageNet-1K benchmark. For experiments on ImageNet-1K itself, models are trained from scratch. For all covariance matrix-based feature pooling methods, we follow the settings in prior works (Li et al., 2017a; 2018; Rahman et al., 2020), where the number of feature channels is reduced to 256 using a $1 \times 1$ convolution for computational efficiency. To ensure consistency with existing methods, input images are resized to $448 \times 448$, and random horizontal flipping is the only data augmentation applied. Models are trained for 100 epochs with an initial learning rate of 0.00012, which is reduced by a factor of 10 at the 45th and 90th epochs on the first five datasets. We set the hyperparameters $K = 4$, $\beta = 0.1$, and $\lambda = 1$ throughout these experiments. For the ImageNet-1K, input images are resized to $224 \times 224$ in accordance with its standard training protocol. The models are trained for 65 epochs from scratch using SGD with an initial learning rate of 0.1, which is decayed by a factor of 10 at the 30th and 45th epochs. Further experimental details can be found in the supplementary materials.

Table 1: Accuracy comparison (top-1%) between three representative covariance matrix based feature pooling methods (iSQRT (Li et al., 2018), iSICE (Rahman et al., 2023), and DeepKSPD (Engin et al., 2018)) and the ones obtained by applying the proposed localized approach. "+ LSOP" denotes the performance is obtained by applying the proposed localized second-order pooling method. The second column represents the types of second-order pooling, and "cov", "pco", and "ker" denote *conventional covariance matrix*, *partial correlation*, and *kernel based non-linear* second-order pooling respectively.

| Backbone | Type | Methods | Birds | Cars | Airplane | Dogs | MIT | Ave Gain |
|---|---|---|---|---|---|---|---|---|
| ResNet-50 | cov | iSQRT | $84.1 \pm 0.4$ | $92.4 \pm 0.3$ | $86.5 \pm 0.5$ | $84.5 \pm 0.3$ | $79.0 \pm 1.1$ | ↑1.4 |
| | | + LSOP (ours) | $\mathbf{86.0 \pm 0.1}$ | $\mathbf{93.3 \pm 0.1}$ | $\mathbf{87.2 \pm 0.2}$ | $\mathbf{86.5 \pm 0.1}$ | $\mathbf{80.7 \pm 0.5}$ | |
| | pco | iSICE | $85.9 \pm 0.4$ | $93.2 \pm 0.2$ | $89.4 \pm 0.2$ | $84.6 \pm 0.2$ | $80.0 \pm 0.3$ | ↑1.7 |
| | | + LSOP (ours) | $\mathbf{87.1 \pm 0.0}$ | $\mathbf{94.5 \pm 0.0}$ | $\mathbf{89.7 \pm 0.2}$ | $\mathbf{86.9 \pm 0.3}$ | $\mathbf{83.3 \pm 0.3}$ | |
| | ker | DeepKSPD | $\mathbf{86.1 \pm 0.2}$ | $92.0 \pm 0.2$ | $83.2 \pm 0.1$ | $87.5 \pm 0.3$ | $82.9 \pm 0.3$ | ↑1.0 |
| | | + LSOP (ours) | $85.9 \pm 0.1$ | $\mathbf{92.2 \pm 0.2}$ | $\mathbf{85.8 \pm 0.1}$ | $\mathbf{88.7 \pm 0.2}$ | $\mathbf{84.0 \pm 0.5}$ | |
| ResNet-101 | cov | iSQRT | $84.7 \pm 0.6$ | $92.9 \pm 0.1$ | $86.7 \pm 0.3$ | $85.1 \pm 0.5$ | $79.0 \pm 0.2$ | ↑1.8 |
| | | + LSOP (ours) | $\mathbf{87.0 \pm 0.1}$ | $\mathbf{94.0 \pm 0.0}$ | $\mathbf{88.8 \pm 0.0}$ | $\mathbf{86.9 \pm 0.1}$ | $\mathbf{80.7 \pm 0.4}$ | |
| | pco | iSICE | $86.7 \pm 0.3$ | $93.7 \pm 0.1$ | $89.8 \pm 0.2$ | $85.3 \pm 0.1$ | $80.5 \pm 0.4$ | ↑1.6 |
| | | + LSOP (ours) | $\mathbf{87.3 \pm 0.1}$ | $\mathbf{94.5 \pm 0.2}$ | $\mathbf{90.1 \pm 0.1}$ | $\mathbf{88.4 \pm 0.2}$ | $\mathbf{83.7 \pm 0.1}$ | |
| | ker | DeepKSPD | $\mathbf{87.1 \pm 0.0}$ | $92.2 \pm 0.4$ | $85.8 \pm 0.2$ | $88.3 \pm 0.1$ | $82.6 \pm 0.5$ | ↑1.0 |
| | | + LSOP (ours) | $86.9 \pm 0.2$ | $\mathbf{92.4 \pm 0.1}$ | $\mathbf{87.7 \pm 0.3}$ | $\mathbf{88.9 \pm 0.2}$ | $\mathbf{85.2 \pm 0.5}$ | |
| Swin-T | cov | iSQRT | $86.2 \pm 0.0$ | $92.0 \pm 0.1$ | $83.2 \pm 0.0$ | $88.8 \pm 0.0$ | $81.2 \pm 1.0$ | ↑1.2 |
| | | + LSOP (ours) | $\mathbf{86.9 \pm 0.0}$ | $\mathbf{92.6 \pm 0.3}$ | $\mathbf{84.9 \pm 0.2}$ | $\mathbf{88.9 \pm 0.1}$ | $\mathbf{84.0 \pm 0.6}$ | |
| | pco | iSICE | $86.9 \pm 0.1$ | $92.7 \pm 0.1$ | $86.7 \pm 0.3$ | $86.9 \pm 0.1$ | $84.3 \pm 0.8$ | ↑0.8 |
| | | + LSOP (ours) | $\mathbf{87.8 \pm 0.1}$ | $\mathbf{93.4 \pm 0.1}$ | $\mathbf{87.5 \pm 0.1}$ | $\mathbf{88.0 \pm 0.1}$ | $\mathbf{84.9 \pm 0.6}$ | |
| | ker | DeepKSPD | $86.8 \pm 0.1$ | $92.2 \pm 0.1$ | $83.9 \pm 0.0$ | $89.2 \pm 0.1$ | $85.9 \pm 0.7$ | ↑0.9 |
| | | + LSOP (ours) | $\mathbf{86.9 \pm 0.1}$ | $\mathbf{92.5 \pm 0.2}$ | $\mathbf{85.8 \pm 0.2}$ | $\mathbf{89.9 \pm 0.1}$ | $\mathbf{87.2 \pm 0.4}$ | |
| DINO-v2 | cov | iSQRT | $86.7 \pm 0.0$ | $92.7 \pm 0.0$ | $74.0 \pm 0.0$ | $88.0 \pm 0.0$ | $91.5 \pm 0.0$ | ↑0.7 |
| | | + LSOP (ours) | $\mathbf{87.4 \pm 0.0}$ | $\mathbf{93.8 \pm 0.0}$ | $\mathbf{75.2 \pm 0.0}$ | $\mathbf{88.2 \pm 0.0}$ | $\mathbf{91.6 \pm 0.0}$ | |
| | pco | iSICE | $88.4 \pm 0.0$ | $93.7 \pm 0.0$ | $\mathbf{79.6 \pm 0.0}$ | $88.8 \pm 0.0$ | $91.2 \pm 0.0$ | ↑0.2 |
| | | + LSOP (ours) | $\mathbf{88.9 \pm 0.0}$ | $\mathbf{94.2 \pm 0.0}$ | $78.4 \pm 0.0$ | $89.1 \pm 0.0$ | $\mathbf{91.9 \pm 0.0}$ | |
| | ker | DeepKSPD | $83.6 \pm 0.0$ | $92.3 \pm 0.0$ | $71.8 \pm 0.0$ | $86.7 \pm 0.0$ | $91.7 \pm 0.0$ | ↑0.4 |
| | | + LSOP (ours) | $\mathbf{83.7 \pm 0.0}$ | $\mathbf{92.3 \pm 0.0}$ | $\mathbf{72.9 \pm 0.0}$ | $\mathbf{87.1 \pm 0.0}$ | $\mathbf{92.1 \pm 0.0}$ | |

Table 2: Impact of feature fusion schemes on computational cost, number of model parameters, inference time per image, and top-1 classification accuracy (%). The symbols "*cro*", "*wsp*", "*gs*", "*con*", and "*saf*" denote cross-attention, weighted sum prediction, gated selection (the fusion scheme used in (Wang et al., 2018)), concatenation, and sample-adaptive feature fusion scheme, respectively.

| Method | Fusion | Birds | Cars | Airplane | Dogs | MIT | MACs (G) | #Para (M) | Latency (ms) | Avg. Top-1 Acc. (%) |
|---|---|---|---|---|---|---|---|---|---|---|
| iSQRT (baseline) | | 84.7 | 92.9 | 86.7 | 85.1 | 79.0 | 8.07 | 49.6 | 10.4 | 85.7 |
| + LSOP | *cro* | 85.8 | 93.1 | 87.9 | 85.2 | 79.7 | 9.04 | 86.2 | 12.7 | 86.3 (↑0.6) |
| | *wsp* | 86.0 | 93.4 | 87.5 | 85.7 | 79.7 | 8.75 | 52.6 | 12.6 | 86.4 (↑0.7) |
| | *gs* | 85.6 | 93.3 | 87.7 | 86.5 | 79.5 | 9.80 | 56.7 | 14.5 | 86.5 (↑0.8) |
| | *con* | 85.7 | 93.4 | 88.2 | **87.1** | **82.5** | 8.77 | 72.3 | 12.6 | 87.4 (↑1.7) |
| | *saf* (ours) | **87.0** | **94.0** | **88.8** | 86.9 | 80.7 | 8.75 | 52.6 | 12.6 | 87.5 (↑**1.8**) |

## 4.2 Comparison with Representative Global Methods

To verify the effectiveness and applicability of the proposed localized method, we select three representative covariance matrix based feature pooling methods, iSQRT (Li et al., 2018), iSICE (Rahman et al., 2023), and DeepKSPD (Engin et al., 2018) as our baselines to make a comparison. iSQRT introduces an iterative way to fast normalize the covariance matrix and this approach has been one of the most widely used approaches in recent covariance matrix pooling related work. As one of the most recent second-order pooling methods,

iSICE proposes a sparse partial correlation algorithm to address the confounding issue and it achieves state-of-the-art performance on many datasets. DeepKSPD utilizes a non-linear kernel to describe the relationship between channels in the training process, that could be regarded as a parametric-based covariance pooling. In this study, we apply our method to all three baseline methods to give a direct and clear comparison to better demonstrate its effect on performance and applicability. In addition, four widely used backbones, i.e., ResNet-50, ResNet-101 (He et al., 2016), Swin-T (Liu et al., 2021), and DINOv2 (Oquab et al., 2024), are utilized to help further verify the applicability of our method on different deep architectures. In particular, the first three backbones are involved in parameter updates during training, while DINOv2 remains entirely frozen throughout the training process to simulate transfer learning from a foundation model. To make a fair comparison, all the baseline methods are re-implemented with the training settings from iSQRT. We also simply follow this setting for our method instead of further parameter tuning to purely demonstrate the impacts coming from the proposed method. For datasets without predefined validation or test splits, we randomly divide the original test set into 20% for validation and 80% for testing. The final results are reported on the test set using the model checkpoint that achieves the best performance on the validation set during training.

### 4.3 Feature Fusion Alternatives

In this subsection, our target is to verify the efficacy of the sample-adaptive feature fusion (referred to as *saf*) scheme. Specifically, we make the study from the following two perspectives. On one hand, we compare *saf* with other four commonly used feature fusion schemes as mentioned in the subsection "Feature Fusion of Local Covariance Matrices", i.e., concatenation (*con*), cross-attention (*cro*), weighted sum prediction (*wsp*), and gated selection (*gs*) by evaluating their classification accuracy. On the other hand, we also report the computational cost in GMACs (Sovrasov, 2018-2024) and inference time with a $224 \times 224$ input image and the number of model parameters, along with the accuracy to provide an overall performance on the trade-off between accuracy and efficiency. For this comparison, iSQRT is used as baseline.

From Table 2, we can see that fusing localized covariance features with all the five mentioned schemes could improve the performance on the five datasets comparing with the baseline. That indicates the proposed attention-based cluster mining effectively benefits the localized second-order pooling no matter what fusion scheme is used. In light of the computational complexity analysis, *saf* shows more advances on keeping good balances between the accuracy and efficiency. As shown, it takes the least extra computational cost (only 8.3% GMac and 5.6% parameters increase) but obtains the most average performance gain (1.8% increase) among the five schemes. This verifies the capability of our framework on both accuracy and efficiency.

The performances are reported in Table 1. In particular, we partition the table into four parts according to the backbone type and provide a individual comparison between the baselines and the ones obtained by applying our method (indicated by "+ LSOP") within each part. An average performance gain on all five datasets is also given in the last column to better demonstrate the impact of our method. Generally, our method obtains better or competitive results on all datasets with all backbone models. In most cases, we could improve the performance by more than 1% for two CNN backbones. In particular, we achieve a performance gain of about 3.3% on MIT with iSICE when using ResNet-50. As to the ViT variants, our method obtains the overall best performance on all datasets. In a nut shell, the performance gain verifies that our method is widely applicable to various backbones and existing covariance matrix based feature pooling methods. This further demonstrates the benefits of conducting localized second-order pooling within the local clusters.

Table 3: Error rate (top-1/top-5%) performance on large-scale classification dataset. † denotes our reimplementation.

| Method | ImageNet-1K | |
|---|---|---|
| | Top-1 $\downarrow$ | Top-5 $\downarrow$ |
| iSQRT (Li et al., 2018)† | 20.97 | 5.73 |
| GM-SOP (Wang et al., 2018)† | 20.42 | 5.31 |
| iSQRT + LSOP (ours) | **20.12** | **5.04** |

### 4.4 On Large-scale Dataset

We further evaluate our method on the ImageNet-1K benchmark. Specifically, we compare it with iSQRT, a representative covariance matrix-based pooling method, and GM-SOP, which shares a similar motivation with our approach. As shown in Table 3, applying LSOP leads to lower top-1 and top-5 error rates compared to the iSQRT baseline. This demonstrates that our method is effective not only on fine-grained datasets but also on large-scale classification tasks. Moreover, our approach outperforms GM-SOP, highlighting the importance of explicit cluster mining in localized second-order pooling frameworks.

Table 4: Accuracy comparison (top-1%) with SOTA FGVC related methods. All the results are quote from the original papers. iSQRT (Li et al., 2018) is selected are the base second-order pooling approach.

| Method | Backbone | Birds | Cars | Airplane |
|---|---|---|---|---|
| TransFG (He et al., 2022) | ViT-B_16 | 91.7 | 94.8 | - |
| IELT (Xu et al., 2023) | ViT-B_16 | 91.8 | - | - |
| FRe-Net (Zhao et al., 2023) | DenseNet-161 | 89.9 | 95.1 | 94.2 |
| FET (Chen et al., 2024) | SwinT-B | 92.9 | 95.9 | 95.3 |
| SwinFG (Ma et al., 2024) | SwinT-B | **93.3** | 96.5 | 95.2 |
| I2-HOFT (Sikdar et al., 2025) | ResNet50+GNN | 91.6 | **96.9** | **96.4** |
| iSQRT+LSOP (ours) | SwintT-B | 91.4 | 95.3 | 95.1 |

### 4.5 Comparison with SOTA FGVC Related Methods

Although fine-grained visual classification (FGVC) is often cited as a representative application motivating second-order pooling, our framework is not specifically tailored for this task. Nevertheless, to comprehensively assess its potential, we conduct additional experiments comparing our method with the state-of-the-art methods designed explicitly for FGVC. To ensure a fair and consistent comparison, we re-implement our LSOP framework strictly following the training settings adopted by existing FGVC state-of-the-art methods, which differ from the settings used in our earlier experiments. Specifically, we evaluate our approach under the same configurations used by TransFG (He et al., 2022), IELT (Xu et al., 2023), FRe-Net (Zhao et al., 2023), FET (Chen et al., 2024), SwinFG (Ma et al., 2024), and I2-HOFT (Sikdar et al., 2025).

As summarized in Table 4, despite not being specifically tailored for FGVC, the proposed LSOP method attains competitive performance on multiple benchmark FGVC datasets when trained using stronger pretrained backbones and the same training protocols as competing methods. In contrast, many existing FGVC approaches are explicitly designed with task-specific mechanisms—such as region-proposal modules, GNN-based relational reasoning, or multi-stage feature extractors—to capture fine-grained cues. These specialized components, while effective for FGVC, often increase architectural complexity and reduce generality. The strong results achieved by our simple and general-purpose framework—without relying on FGVC-specific designs—further validate both the effectiveness and the applicability of LSOP for visual recognition tasks.

### 4.6 Ablation Study

In this section, we conduct ablation studies to investigate several main properties of our method. In particular, we first demonstrate the impact of two key hyperparameter used in our framework, which are $\beta$ in eq. (5) and the number of cluster $K$. Moreover, we provide more studies including the accuracy and computational overhead against the number of clusters, detail analysis on the sample-adaptive feature fusion scheme, more visual examples with attention maps in the supplementary material.

#### 4.6.1 On the Number of Clusters $K$

As seen in Figure 2, while the results do not exhibit a consistent or monotonic trend, we observe that extreme settings—either too few or too many clusters—tend to yield slightly inferior performance. In contrast, moderate cluster numbers generally lead to better results. Nevertheless, all settings consistently outperform the baseline by a clear margin, indicating that our method is robust to the choice of cluster granularity.

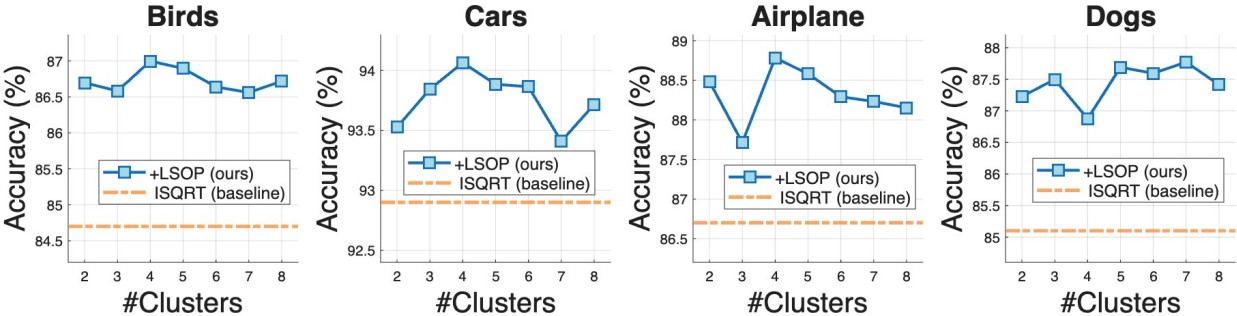

Figure 2: Impact of the number of clusters on classification performance across four FGVC datasets.

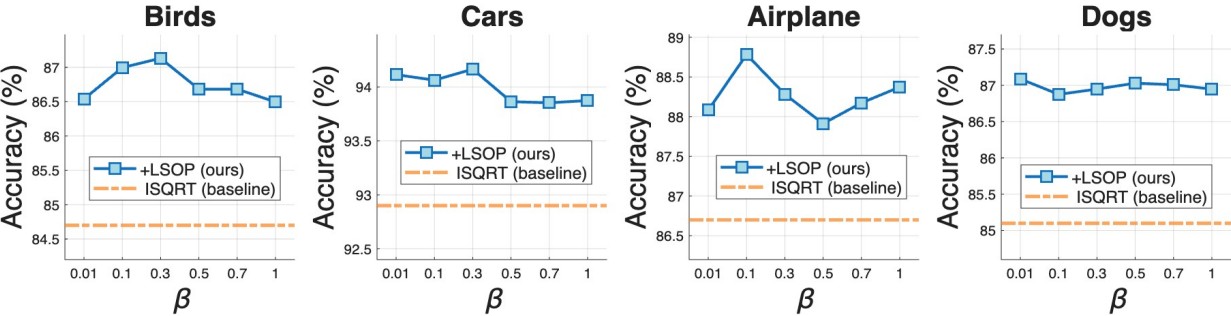

Figure 3: Impact of regularization coefficient $\beta$ on classification performance across four FGVC datasets.

### 4.6.2  On the Selection of $\beta$

A similar observation can be made based on the results shown in Figure 3. Although no clear trend emerges as the regularization coefficient $\beta$ varies, our method consistently outperforms the baseline by a substantial margin across a wide range of $\beta$ values. This further confirms the robustness of our approach with respect to the choice of the regularization strength.

## 5  Conclusion

In this paper, we present an efficient end-to-end framework for localized second-order pooling, where covariance is computed within explicitly mined clusters. By integrating a dedicated cluster-mining branch with a sample-adaptive feature fusion strategy, our method overcomes key limitations of existing local second-order approaches. Experiments on multiple fine-grained and large-scale benchmarks show that LSOP achieves strong accuracy–efficiency trade-offs while outperforming prior methods. As future work, we aim to extend the proposed localized pooling mechanism to intermediate network stages.

## Acknowledgments

Zhongyan Zhang and Lei Wang were supported by the Australian Research Council via the Discovery Project with grant number DP200101289. Saimunur Rahman was supported by the CSIRO's Machine Learning and Artificial Intelligence (MLAI) FSP and CSIRO's Data61 Science Digital. Piotr Koniusz was supported by the UNSW Merit and the CSIRO Allocation Scheme. This research was undertaken with the assistance of resources and services from the National Computational Infrastructure (NCI), which is supported by the Australian Government.

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

## A  Datasets

In this section, the description of the six datasets, Birds (Wah et al., 2011), Cars (Krause et al., 2013), Airplane (Maji et al., 2013), Dogs (Dataset, 2011), MIT (Quattoni & Torralba, 2009), and ImageNet-1K (Deng et al., 2009) used in the experiment are provided. Some example images randomly sampled from these datasets are shown in Figure 4.

**Birds.** As one of the most popular datasets for fine-grained visual classification tasks, it contains 11,788 images of 200 classes in total. The main difficulty lies in the subtle differences between bird species which are hard to distinguish by even human experts. This dataset provides the bounding box annotations for ground truth objects. However, we did not use this information at all.

**Cars.** This dataset has a total of 16,185 images of 196 car classes. The classes are categories concerning car production year, car model, and car manufacturer. Compared with the Birds dataset, the car images in this dataset have a more complex background.

**Airplane.** Similar to the above two datasets, the airplane dataset is also a widely used fine-grained visual classification dataset. It has 10,000 images in total coming from 100 aircraft classes. These images are uniformly assigned to the 100 classes.

**Dogs.** The Stanford Dogs dataset contains 20,580 images of 120 breeds of dogs from around the world. This dataset has been built using images and annotation from ImageNet for the task of fine-grained image categorization.

**MIT.** MIT is an indoor scene recognition dataset that has a total of 15,620 images from 67 classes. The main difficulty is that while some indoor scenes (*e.g.*, corridors) can be well characterized by global spatial properties, others (*e.g.*, bookstores) are better characterized by the objects they contain.

**ImageNet-1K.** ImageNet-1K is a landmark dataset with over 1.2 million images across 1,000 categories, used extensively for benchmarking image classification algorithms in deep learning. It has been instrumental in the evolution of convolutional neural networks by providing a diverse and challenging set of classes for models to learn from.

## B  Computational Overhead w.r.t. the Number of Clusters

We demonstrate the impact of the number of clusters on the computational overhead by applying our method with a set of different cluster numbers (i.e., 2, 3, 4, 5, 6, 7, and 8). As seen in Figure 5, as the number of clusters increases, both the computational complexity evaluated by GMac (on the left side) and the number of parameters (on the right side) increases accordingly. This is expected since the proposed method needs to conduct the matrix normalization process multiple times corresponding to the number of local covariance

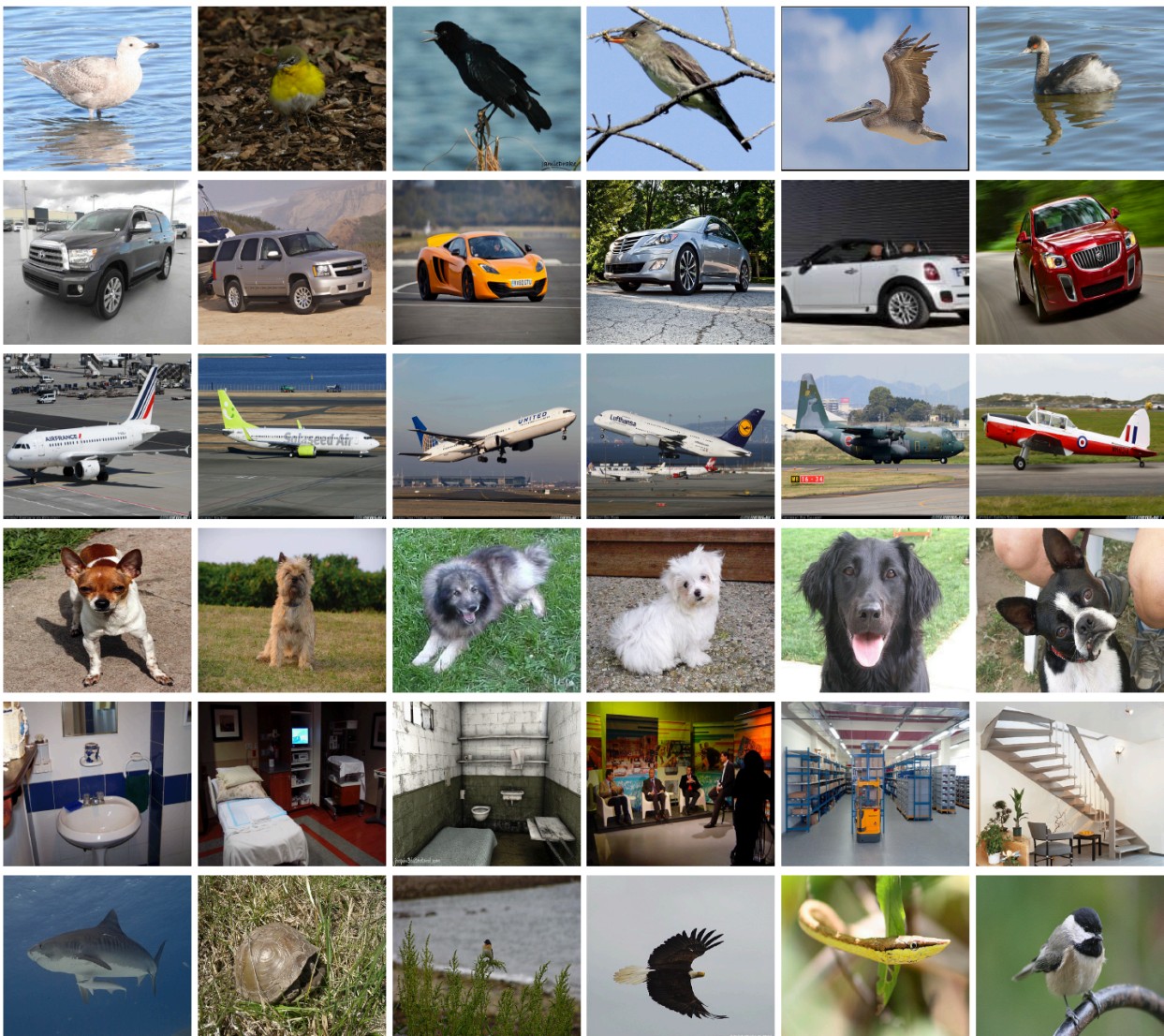

Figure 4: Images sampled from the datasets used in our experiments. Rows 1, 2, 3, 4, 5, and 6 have images from Birds, Cars, Airplane, Dogs, MIT, and ImageNet-1K, respectively.

matrices. However, compared to the three other feature fusion schemes which are concatenation (denoted as "con"), cross-attention (denoted as "cro"), and gated selection (denoted as "gs"), the proposed sample-adaptive feature fusion dramatically saves the extra computational cost on both complexity and the number of parameters perspectives[2].

## C    Illustration of Local Descriptor Clusters on Example Images

We provide some visual examples of the attention maps generated by our attention-based cluster mining block. As shown in Figure 6, 7, and 8, the first column displays the raw images and the following columns provide the attention maps calculated with the soft assignment weights in eq. (1) in the main text. In particular, the position in red denotes the value there is high and the position in blue denotes the opposite. In addition, the corresponding weight for each cluster $(1/\tau)$ is listed at the bottom of each column. As seen,

---

[2]Note the weighted sum prediction (denoted as "wsp" in the main text) is not displayed since it shares the same computational complexity with our fusion scheme.

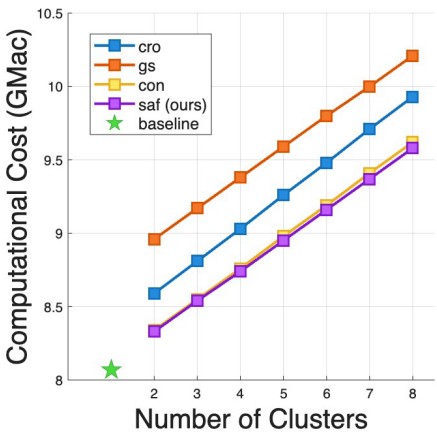 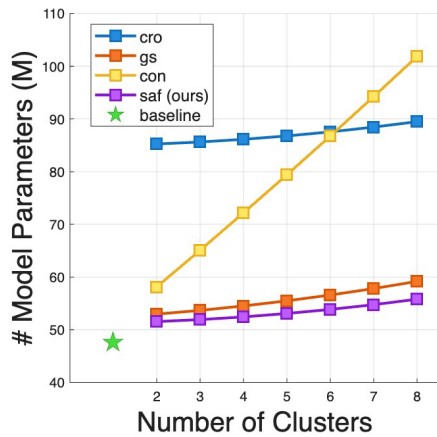

Figure 5: Computational overhead vs. the number of clusters when conducting iSQRT (Li et al., 2018) with ResNet101. The "baseline" denotes the original iSQRT.

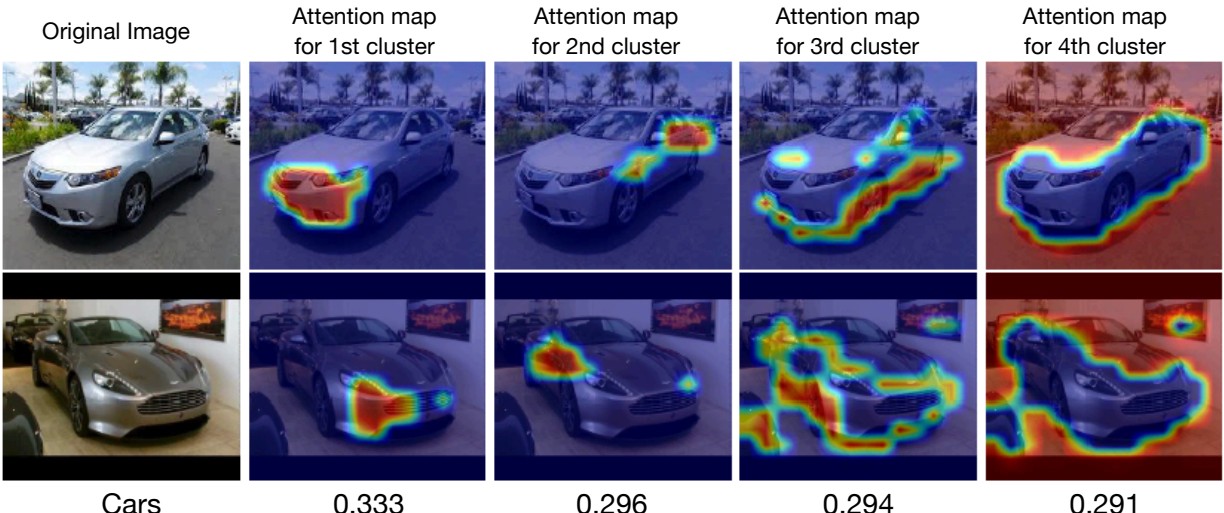

Figure 6: Visualization of the end-to-end learned clusters ($K$=4) on images randomly sampled from the Cars dataset. The first column shows the original images, while the following four columns present the attention maps corresponding to the first, second, third, and fourth clusters, respectively.

the attention maps corresponding to the same cluster (presented in the same column) consistently focus on the same semantic pattern across different images within a dataset. For example, the first cluster consistently pays more attention to the front of the cars and the last cluster usually focuses more on the background no matter what car types are in the images in the Cars dataset (first line in Figure 6). The same result could be observed from the images of the other datasets. This result demonstrates that our attention-based cluster mining branch effectively mines the underlying semantic information and generates reliable clusters for the dataset.

## D    Further Discussion on Impact of Number of Cluster

The four datasets have quite different number of classes, specifically 200 in Birds, 196 in Cars, 100 in Airplane, and 120 in Dogs. Nevertheless, as seen in Figure 9, the $K$ value corresponding to the highest classification accuracy is 16, 24, 4 and 24 for Birds, Cars, Airplane, and Dogs respectively. This result

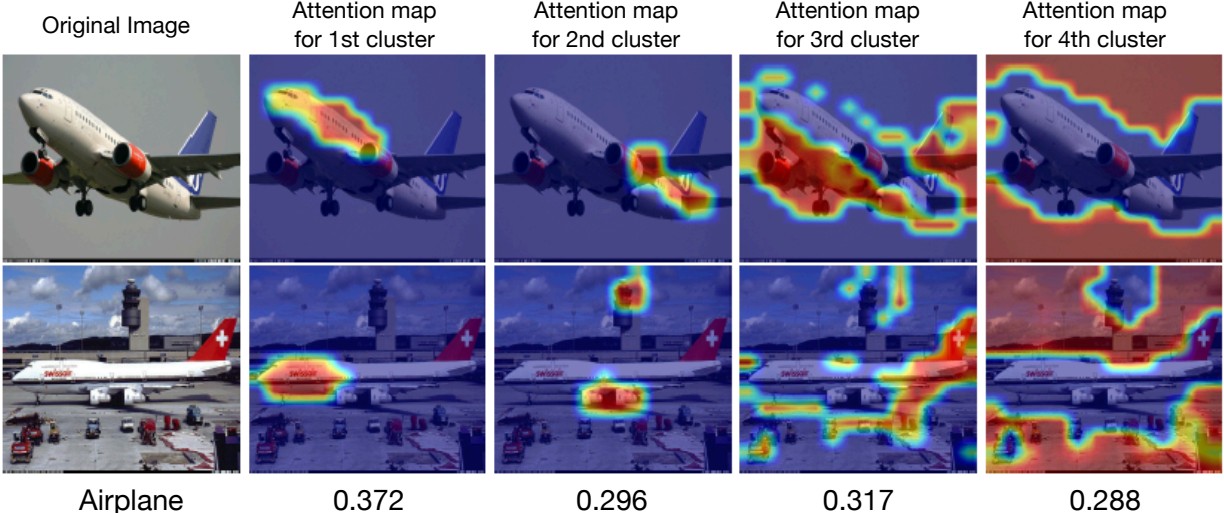

Figure 7: Visualization of the end-to-end learned clusters ($K$=4) on images randomly sampled from the Airplane dataset. The first column shows the original images, while the following four columns present the attention maps corresponding to the first, second, third, and fourth clusters, respectively.

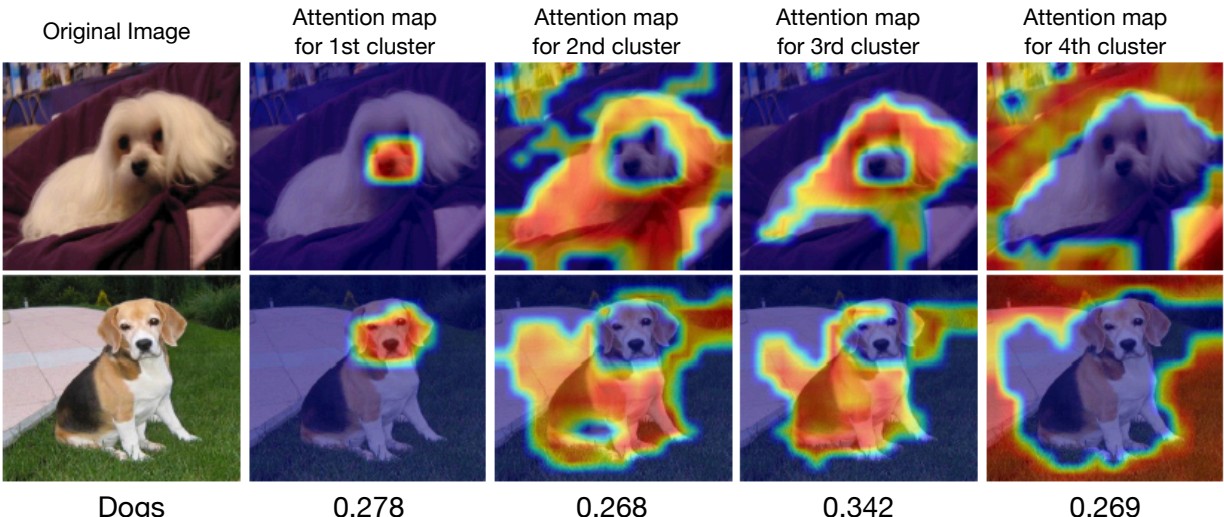

Figure 8: Visualization of the end-to-end learned clusters ($K$=4) on images randomly sampled from the Dogs dataset. The first column shows the original images, while the following four columns present the attention maps corresponding to the first, second, third, and fourth clusters, respectively.

indicates that the optimal $K$ does not necessarily scale with the number of classes in a dataset. When considering the geometric complexity of objects, comparisons can be made across the Birds, Dogs, Cars, and Airplane datasets. The main objects in Birds and Dogs datasets are animals, which exhibit high geometric complexity. In contrast, vehicles and aircraft in the Cars and Airplane datasets possess relatively lower geometric complexity. As illustrated in the figure, with varying values of $K$, no distinct monotonic trend can be observed in the final classification performance across all four datasets. This observation holds consistently for both higher-complexity datasets (Birds, Dogs) and lower-complexity datasets (Cars, Airplane). Across all these conditions, a key strength of our method emerges: Our model consistently outperforms the baseline

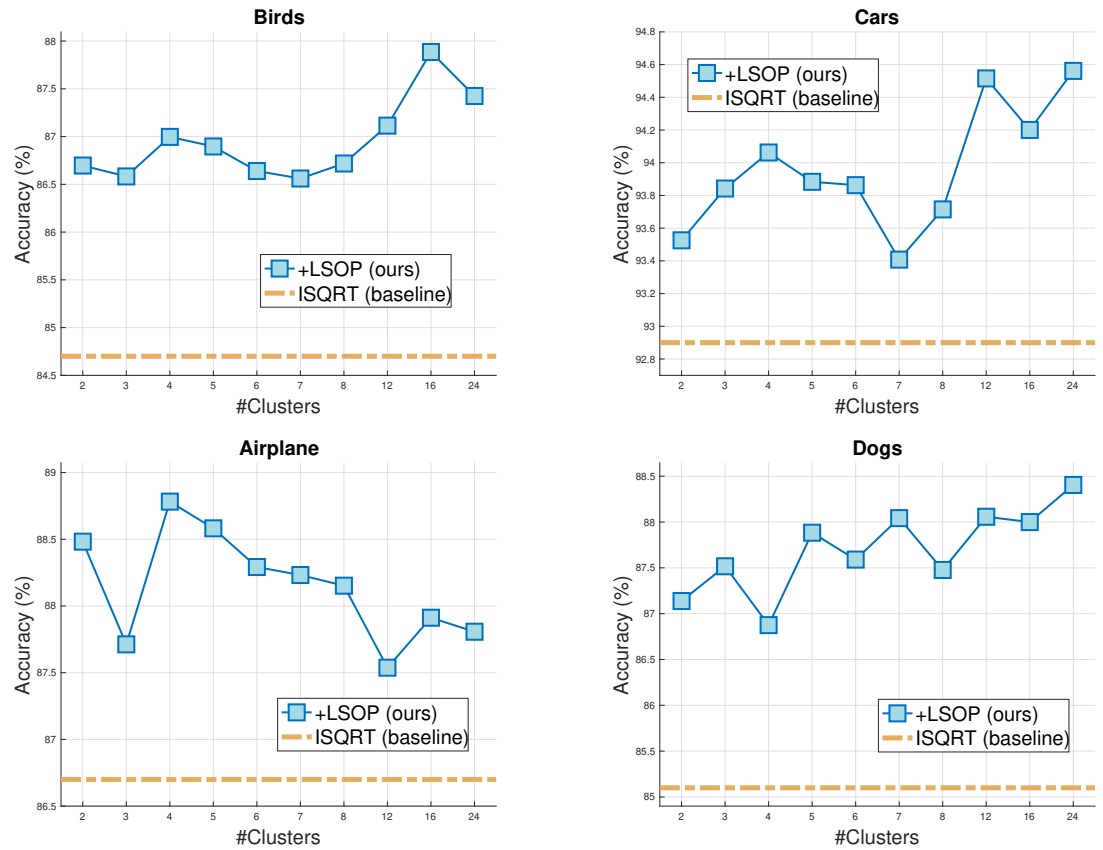

Figure 9: Impact of the number of clusters on classification performance.

whose performance is denoted by the yellow dashed line, regardless of the value of $K$ in a reasonably large range. This suggests the insensitivity of our model to $K$. This desirable property is not coincidental; rather, this insensitivity to the choice of $K$ stems from our proposed sample-adaptive feature fusion module. This module can automatically conduct information compensation and trade-off among multiple branches according to distinct branch-wise features. Consequently, even under substantial variations in $K$, the final classification performance is only marginally affected.

# E   Visualization on the Clustering

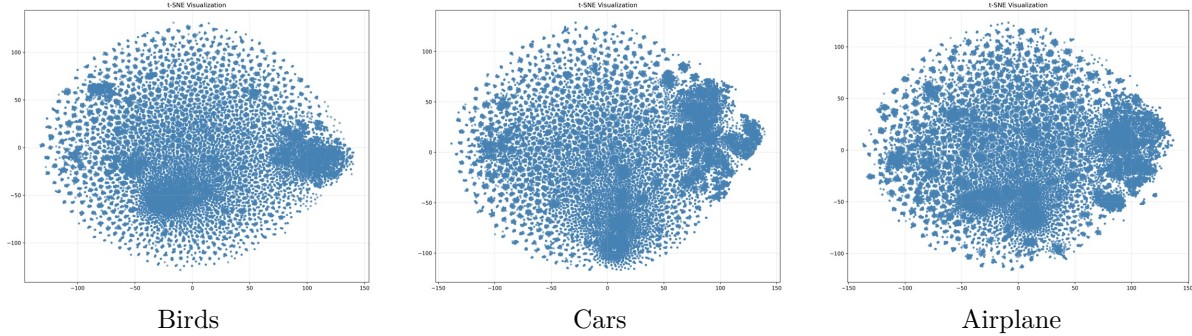

Figure 10: *t*-SNE for dataset Birds, Cars, and Airplane.

To verify the motivation of using localized covariance matrices in our framework, we conduct the following studies. In particular, we train the iSQRT model, which is based on a single global covariance matrix, on a fine-grained dataset. The trained model is then used to extract local descriptors from images and their data distribution is visualized via $t$-SNE. In total, three datasets, namely Birds, Cars, and Airplane, are tested. For each of them, we perform uniformly random sampling across all categories. A total of $20,000$ instances are sampled to generate $t$-SNE visualization plots. As illustrated in Figure 10, the distributions of local feature descriptors do not follow a unimodal distribution. Instead, the descriptors aggregate into multiple overlapping clusters. This result demonstrates that the unimodality assumption does not hold and supports our use of multiple localized covariance matrices.

