# OpenReview forum: "An Efficient End-to-End Framework for Localized Second-Order Pooling"
_TMLR — Accepted by TMLR_

### Review · Reviewer_tPSv · 2026-03-10

**Summary Of Contributions:**

This paper studies an important problem of deep image classification. The authors propose an efficient end-to-end framework for localized sceond-order pooling. Generally, the paper is well-written and easy to follow. Extensive experiments show the effectiveness of the proposals.

**Audience:**

Yes

**Audience Explanation:**

The computer vision community will be interested in this paper.

**Claims And Evidence:**

Yes

**Claims Explanation:**

The claims are supported via extensive experiments.

**Requested Changes:**

1. It seems that the second-order pooling is time-consuming. It would be better to include more experiments regarding efficiency, e.g., comparing training time, inference time, and FLOPs over baselines.
2. It is suggested to include the theoretical computational complexity analysis in the manuscript.
3. It would be better to include Pseudocode for the scecond-order pooling mechanism for better understanding.

---

> ### Author Response · Authors · 2026-04-24
> **Thank you for your valuable comments. We have put all the answers in text form here and add the necessary figs in the Appendix in the main text. Please go to the revised main text for figure demonstrations.**
>
> Caption: Comparison on training time and FLOPs for different feature fusion approaches.
>
> Method | Fusion | FLOPs (GMac) | Training Time (s / epoch)
>
> iSQRT | -  | 8.07 | 60.3
>
> -  |  cro | 9.04 | 64.5
>
> -  | wsp | 8.75 | 63.1
>
> -  | gs | 9.80 | 68.2
>
> -  | con | 8.77 | 67.8
>
>  + LSOP | saf (ours) | 8.75 | 63.6
>
> A1: We appreciate your valuable suggestions. We have presented the inference time, denoted as Latency, in the second last column of Table 2 in the main text. On this basis, we supplement the training time and floating-point operations (FLOPs) of different feature fusion strategies in the table herein. As demonstrated in Table~\ref{tab:time}, compared with the baseline iSQRT model, the proposed method does not introduce substantial additional computational time overhead. Furthermore, in contrast to other feature fusion approaches, our method achieves the optimal overall performance.
>
> A2: Thank you. The main time-consuming components of the proposed LSOP framework are the localized covariance matrix (as demonstrated in Eq.(3) in the main text) and covariance normalization modules. Other components, including online clustering, and sample-adaptive feature fusion operations only incur low-order computational overhead and are not the main time-consuming parts. For the localized covariance matrix module, its complexity is determined by parallel K-way computation. For each single branch, the core computation is the matrix multiplication between the feature matrix and the weighted centering matrix, with a complexity of $\mathcal{O}(d^2 M)$, where $d$ denotes the dimensionality of the reduced feature and $M = h \times w$ ($h$ and $w$ are the height and width of the feature map, respectively). With $K$ parallel branches ($K$ is the number of clusters), the overall complexity of this module is $\mathcal{O}(K d^2 M)$, corresponding to the computation of the weighted covariance matrix for each cluster.
>
> For the covariance normalization module, the core computation is matrix square rooting or sparse matrix optimization on $d \times d$ matrices. For this part, its computational complexity is exactly the same as that of iSQRT). And this is the dominant time-consuming component of the LSOP framework.
>
> A3: Given that no explicit constraint is enforced on the specific formulation of second-order pooling within our framework, we elaborate the pseudocode of the localized covariance matrix computation below:
>
> \KwIn{$x \in \mathbb{R}^{B \times D_{in} \times H \times W}$}
>
> \KwOut{$z \in \mathbb{R}^{B \times D_{out}}$}
>
> $A \leftarrow \text{OnlineCluster}(x)$ \# Extract features and attention \;
>
> $x \leftarrow \text{ConvDR}(x)$        \# Reduce feature dimension \;
>
> $\Sigma \leftarrow \text{WeightedCovPool}(x, A)$ \# Compute weighted covariance \;
>
> $\Sigma \leftarrow \text{iSQRT}(\Sigma, T)$ \# Matrix square rooting \;
>
> $V \leftarrow \text{TriuVec}(\Sigma)$  \# Vectorize upper triangular part \;
>
> $w \leftarrow \text{Predictor}(x)$ \# Predict token weights \;
>
> $z \leftarrow \sum w_k V_k$  \# Fuse features with weights \;
>
> \Return $z$ \;

---

### Review · Reviewer_gVzA · 2026-03-18

**Summary Of Contributions:**

This paper proposes an improved classification training method and model architecture based on second-order pooling, which represents an image as a set of covariance matrices of local feature descriptors.
The paper argues that representing an image with a single global covariance matrix is suboptimal, and it should be computed for each local descriptor cluster and then combined.
The proposed method has an end-to-end clustering and feature-learning framework that uses a slot attention module.
Experimental results show that the proposed method improved accuracy on image classification datasets.

**Audience:**

Yes

**Audience Explanation:**

Improving the architecture of image classification models is beneficial to practitioners.

**Broader Impact Concerns:**

There is no broader impact statement.

**Claims And Evidence:**

No

**Claims Explanation:**

Please refer to the requested changes.

**Requested Changes:**

### Major
- Although the paper argues that computing a single global covariance matrix by assuming the local feature descriptors' unimodality is suboptimal, whether the unimodality actually breaks should be verified, since this is an important motivation of this study. For example, visualizing local descriptors in existing models that use a single global covariance matrix via t-SNE or UMAP would be interesting.
- Visualizing cluster importance $1/\tau\_k$ with the cluster visualizations in Figs. 6-8 would be interesting if we can observe that semantically important regions have large weights.
- It would be more convincing to verify that the clustering by slot attention contributes to improvements. For example, if the clustering quality (e.g., silhouette score, mutual information) and the resulting accuracy or loss correlate strongly (e.g., by varying $K$), it would demonstrate the importance of clustering.
- It would be more persuasive to examine the effect of second-order pooling. For example, replacing the flattened covariance matrix $h(\Sigma\_k)$ in Eq. (5) with the simple mean vector $\hat{\mu}\_k$ in Eq. (3) could be a verification of that.


### Minor
- Reference to a figure is broken in Appendix C (p. 17).
- $\theta$ in Eq. (5) is undefined; is it the same as the classifier weight $\omega$ in Eq. (2)?

---

> ### Author Response · Authors · 2026-04-24
> **Thank you for your valuable comments. We have put all the answers in text form here and add the necessary figs in the Appendix in the main text. Please go to the revised main text for figure demonstrations.**
>
> A1: Thank you for the suggestion. To address it, we train the iSQRT model, which is based on a single global covariance matrix, on a fine-grained dataset. The trained model is then used to extract local descriptors from images and their data distribution is visualized via t-SNE. In total, three datasets, namely Birds, Cars, and Airplane, are tested. For each of them, we perform uniformly random sampling across all categories. A total of 20,000 instances are sampled to generate t-SNE visualization plots.
>
> As illustrated in Figure 10 in Section E in the Appendix, the distributions of local feature descriptors do not follow a unimodal distribution. Instead, the descriptors aggregate into multiple overlapping clusters. This result demonstrates that the unimodality assumption actually breaks and supports our use of multiple localized covariance matrices.
>
> A2: We statistically calculated the weight $1/\tau$ corresponding to each cluster and incorporated all quantitative results into Figures 6–8 in the main text. In particular, the value of the weights for Cars, Airplane, and Dogs are [0.333, 0.296, 0.294, 0.291], [0.372, 0.296, 0.317, 0.288], and [0.278, 0.268, 0.342, 0.269] respectively. As illustrated in these figures:
>
> For the Cars dataset, the weight values of the front logo area of vehicles are relatively higher (i.e., $3.33$), while those of the background regions remain low (i.e., 0.291);
>
> For the Airplane dataset, the nose section of aircraft exhibits dominant weights (i.e., 0.372), with the background yielding the lowest weight (i.e., 0.288);
>
> In the Dogs dataset, higher weights are assigned to the body regions of dogs (i.e., 0.342), whereas the background areas present comparatively weak weight responses (i.e., 0.269).
>
> This suggests that the sample-adaptive feature fusion module can automatically adjust the information weights assigned to different clusters and implement compensatory adjustments accordingly.
>
> A3: We would like to mention that our framework is a fully end-to-end feature learning paradigm. Consequently, different choices of the hyperparameter $K$ will yield distinct feature extraction models, which further lead to distinct feature spaces. This discrepancy makes it awkward to directly compare the clustering performance under varying K settings (e.g., by silhouette score). To answer this question, we adopt an alternative scheme to check the correlation between clustering effects and final classification results. We set different values of $K$ (ranging from 2 to 5) and adopt t-SNE (as shown in Figure 11 in the Appendix) on the local descriptors to qualitatively evaluate the clustering performance on the Birds dataset. As seen, there is no explicit and direct correlation between clustering quality and classification performance.
>
> A4: As suggested, we replaced the flattened covariance matrix in Eq.(5) with the mean vector $\hat{{\mu}}_k$, while keeping all other experimental settings unchanged. The results are referred as $\mu$+LSOP in Table 1. Also, iSQRT which is the base second-order pooling method is included as a reference.
>
> Caption: Top-1 accuracy (\%) comparison with SOTA FGVC-related methods. Their results are quoted from the original papers.
>
> Method | Birds | Cars | Airplane | Dogs | MIT | Avg
>
> iSQRT   | 84.7 $\pm$ 0.6 | 92.9 $\pm$ 0.1 | 86.7 $\pm$ 0.3 | 85.1 $\pm$ 0.5 | 79.0 $\pm$ 0.2 | 85.7
>
> $\mu$+LSOP | 85.0 $\pm$ 0.1 | 93.0 $\pm$ 0.2 | 86.2 $\pm$ 0.1 | \textbf{87.8} $\pm$ 0.1 | \textbf{83.7} $\pm$ 0.0 | 87.1
>
> iSQRT+LSOP (ours) | \textbf{87.0} $\pm$ 0.1 | \textbf{94.0} $\pm$ 0.0 | \textbf{88.8} $\pm$ 0.0 | 86.9 $\pm$ 0.1 | 83.5 $\pm$ 0.1 | \textbf{88.0}
>
> As shown in Table 1, by comparing the 2nd and 3rd rows,  it can be seen that our method iSQRT+LSOP achieves higher average accuracy ($88.0$ vs. $87.1)$ than $\mu$+LSOP. Specifically, it outperforms on the three fine-grained image datasets (i.e., Birds, Cars, and Airplane) although slightly inferior to it on Dogs and MIT. Also, comparing the 1st and 2nd rows shows that $\mu$+LSOP even achieves higher average accuracy than iSQRT. This indicates the benefits of the localization idea and the feature fusion strategy in our work. Together, the results in Table 1 shows that the effect of second-order pooling becomes more pronounced through our work.
>
> A5: Thank you. We have revised that part in the main text and highlighted it in blue.
>
> A6: Yes, we have revised it to make it consistent with Eq.(2) and highlighted in blue in the main text.

---

### Review · Reviewer_6tR9 · 2026-03-22

**Summary Of Contributions:**

The paper addresses a recurring bottleneck in fine-grained visual recognition i.e. the limitation of global second-order pooling (SOP). While SOP is excellent at capturing complex feature correlations, doing so globally often washes out the multi-modal distributions of local features. To solve this, the authors introduce something called localized second-order pooling (LSOP) which is posed as an efficient, end-to-end framework. The framework addresses the following:
1. Utilizes a slot attention mechanism to learn semantically coherent clusters within the feature space. This is a clever shift from static partitioning to a learnable, data-driven approach.
2. By applying soft asignments, the model computes weighted covariance matrices for these discovered clusters, capturing more granular local statistics.
3. A lightweight sample-adaptive feature fusion scheme that predicts scaling factors per instance.
4. The authors have shown through various experiments that LSOP is highly modular and improves performance across various backbones and pooling baselines.

**Additional Comments:**

I think the modularity of LSOP is its strongest selling point. Being able to "plug and play" this framework with various backbones and existing second-order methods makes it a very practical contribution. The integration of slot attention is particularly elegant.

**Audience:**

Yes

**Audience Explanation:**

Audience interested in representation learning and computer vision will find value in how this paper bridges classic second-order pooling methods with modern attention mechanisms. The use of slot attention for feature space partitioning is a creative design choice that has implications beyond just pooling.

**Broader Impact Concerns:**

None.

**Claims And Evidence:**

Yes

**Claims Explanation:**

The evidence provided in the paper is thorough covering both performance and efficiency. LSOP consistently boosts accuracy across five datasets regardless of whether the backbone is a CNN or a transformer. The sample-adaptive feature fusion scheme is also validated well in Table 2. It is refreshing to see a paper that doesn't just conduct studies on improving SOTA but also accounts for MACs, parameter counts and actual latency. Also, the cluster visualizations (figures 6-8) demonstrate that the attention mechanism is actually considering semantically meaningful parts, which validates the underlying intuition of the paper.

**Requested Changes:**

To make the paper even stronger, I'd like to see the authors address the following points:
1. It would be helpful to see a more detailed discussion on the choice of number of clusters. Does the optimal "K" scale with the number of classes in the dataset or is it more tied to the geometric complexity of the objects? This potentially could be a follow-up contribution in itself but it would be good to see it addressed in some way.
2. A quick pass through equations 1-5 to ensure total consistency in the loss function notation would improve readability for those looking to implement this.

---

> ### Author Response · Authors · 2026-04-24
> **Thank you for your valuable comments. We have put all the answers in text form here and add the necessary figs in the Appendix in the main text. Please go to the revised main text for figure demonstrations.**
>
> A1: Thank you for your valuable comments. As suggested, we investigate the relationship between the number of clusters, $K$, and the classification accuracy on all the four fine-grained image datasets, as shown in Figure 2 in the main text. In addition, to further investigate the impact of number of cluster, we extend this experiment and add the results in Figure 9 in Section D in the Appendix. Based on the results in this figure, we conduct the following discussion:
>
> 1) The four datasets have quite different number of classes, specifically $200$ in Birds, $196$ in Cars, $100$ in Airplane, and $120$ in Dogs. Nevertheless, as seen in Figure 2, the $K$ value corresponding to the highest classification accuracy is $16$, $24$, $4$ and $24$ for Birds, Cars, Airplane, and Dogs respectively. This result indicates that the optimal $K$ does not necessarily scale with the number of classes in a dataset.
>
> 2) When considering the geometric complexity of objects, comparisons can be made across the Birds, Dogs, Cars, and Airplane datasets. The main objects in Birds and Dogs datasets are animals, which exhibit high geometric complexity. In contrast, vehicles and aircraft in the Cars and Airplane datasets possess relatively lower geometric complexity. As illustrated in the figure, with varying values of $K$, no distinct monotonic trend can be observed in the final classification performance across all four datasets. This observation holds consistently for both higher-complexity datasets (Birds, Dogs) and low-complexity datasets (Cars, Airplane).
>
> 3) Our model consistently outperforms the baseline whose performance is denoted by the yellow dashed line, regardless the value of K in a reasonably large range. This suggests the insensitivity of our model to $K$.
>
> 4) This insensitivity to the choice of $K$ stem from our proposed sample-adaptive feature fusion module. This module can automatically conduct information compensation and trade-off among multiple branches according to distinct branch-wise features. Consequently, even under substantial variations in K, the final classification performance is only marginally affected.
>
> A2: Thank you. We have revised the inconsistent notations in the equations accordingly and highlighted them in blue in the main text.

---

### Decision · Action_Editor_SvY5 · 2026-05-05

**Recommendation:** Accept as is

**Audience:**

Yes

**Audience Explanation:**

Computer vision, image classification / recognition, and feature learning communities will be interested in the findings of this paper.

**Claims And Evidence:**

Yes

**Claims Explanation:**

The authors support all claims (regarding both the motivation / problem the proposed method is tackling and the effectiveness of the proposed method in practice) through extensive experiments post-rebuttal.